# Intermediate Repeat Expansion in the *ATXN2* Gene as a Risk Factor in the ALS and FTD Spanish Population

**DOI:** 10.3390/biomedicines12020356

**Published:** 2024-02-02

**Authors:** Daniel Borrego-Hernández, Juan Francisco Vázquez-Costa, Raúl Domínguez-Rubio, Laura Expósito-Blázquez, Elena Aller, Ariadna Padró-Miquel, Pilar García-Casanova, María J. Colomina, Cristina Martín-Arriscado, Rosario Osta, Pilar Cordero-Vázquez, Jesús Esteban-Pérez, Mónica Povedano-Panadés, Alberto García-Redondo

**Affiliations:** 1ALS Research Laboratory Unit, Department of Neurology, Hospital Universitario 12 de Octubre, 28041 Madrid, Spain; lexpositobla.imas12@h12o.es (L.E.-B.); mariadelpilar.cordero@salud.madrid.org (P.C.-V.); jesteban.hdoc@salud.madrid.org (J.E.-P.); ela@h12o.es (A.G.-R.); 2Neuromuscular Unit, ERN-NMD Group, Department of Neurology, Hospital Universitario y Politécnico La Fe, Instituto de Investigación Sanitaria La Fe, 46026 Valencia, Spain; juan.vazquez@uv.es (J.F.V.-C.); garciacasanovapilar@gmail.com (P.G.-C.); 3Centro de Investigación Biomédica en Red de Enfermedades Raras (CIBERER), 28029 Madrid, Spain; aller_ele@gva.es; 4Department of Medicine, University of Valencia, 46010 Valencia, Spain; 5Motoneuron Functional Unit, Hospital Universitari de Bellvitge, 08907 L’Hospitalet de Llobregat, Spain; rdominguez@bellvitgehospital.cat (R.D.-R.); mpovedano@bellvitgehospital.cat (M.P.-P.); 6Genetics Department, Hospital Universitario y Politécnico La Fe, Instituto de Investigación Sanitaria La Fe, 46026 Valencia, Spain; 7Genetics Laboratory (LCTMS), Bellvitge University Hospital-IDIBELL, 08908 L’Hospitalet de Llobregat, Spain; apadro@bellvitgehospital.cat; 8Anesthesia Service Unit, Hospital Universitari de Bellvitge, 08907 L’Hospitalet de Llobregat, Spain; mjcolomina@gmail.com; 9Clinical Research Unit, Hospital Universitario 12 de Octubre, 28041 Madrid, Spain; cmartinarriscado.imas12@h12o.es; 10Laboratório de Genética e Biotecnologia (LAGENBIO), Centro de Investigación Biomédica en Red de Enfermedades Neurodegenerativas (CIBERNED), Aragon Institute for Health Research (IIS Aragon), Zaragoza University, 50013 Zaragoza, Spain; osta@unizar.es

**Keywords:** amyotrophic lateral sclerosis, *ATXN2*, risk factor, frontotemporal dementia, mutation, poli-Q expansion

## Abstract

Intermediate CAG expansions in the gene ataxin-2 (*ATXN2*) are a known risk factor for ALS, but little is known about their role in FTD risk. Moreover, their contribution to the risk and phenotype of patients might vary in populations with different genetic backgrounds. The aim of this study was to assess the relationship of intermediate CAG expansions in *ATXN2* with the risk and phenotype of ALS and FTD in the Spanish population. Repeat-primed PCR was performed in 620 ALS and 137 FTD patients in three referral centers in Spain to determine the exact number of CAG repeats. In our cohort, ≥27 CAG repeats in *ATXN2* were associated with a higher risk of developing ALS (odds ratio [OR] = 2.666 [1.471–4.882]; *p* = 0.0013) but not FTD (odds ratio [OR] = 1.446 [0.558–3.574]; *p* = 0.44). Moreover, ALS patients with ≥27 CAG repeats in *ATXN2* showed a shorter survival rate compared to those with <27 repeats (hazard ratio [HR] 1.74 [1.18, 2.56], *p* = 0.005), more frequent limb onset (odds ratio [OR] = 2.34 [1.093–4.936]; *p* = 0.028) and a family history of ALS (odds ratio [OR] = 2.538 [1.375–4.634]; *p* = 0.002). Intermediate CAG expansions of ≥27 repeats in *ATXN2* are associated with ALS risk but not with FTD in the Spanish population. ALS patients carrying an intermediate expansion in *ATXN2* show more frequent limb onset but a worse prognosis than those without expansions. In patients carrying *C9orf72* expansions, the intermediate *ATXN2* expansion might increase the penetrance and modify the phenotype.

## 1. Introduction

Amyotrophic lateral sclerosis (ALS) is a progressive adult-onset neurodegenerative disease affecting both upper (UMN) and lower motor neurons (LMN) [1,2]. Disease onset is usually focal, but it spreads to other body regions, causing weakness, muscle atrophy, cramps, brisk reflexes, dyspnoea and dysphagia, finally leading to death due to respiratory failure [3,4]. Sixty five percent of ALS patients have limb onset, thirty percent have bulbar onset and five percent have other phenotypes [5].

Over 10% of ALS cases report a family history of ALS or frontotemporal dementia (FTD), considered as familial ALS (FALS) [6], while those without any family history are called sporadic ALS (SALS). Remarkably, both FALS and SALS patients largely share genetic causes, and mendelian mutations can be found in up to 10% of SALS cases. Mendelian mutations causing ALS show considerable founders effects, resulting in a variable genetic background even among populations of European ancestry [7]. Moreover, aggregation studies have identified an overlap between ALS and other neuropsychiatric diseases [8].

FTD is an insidious neurodegenerative syndrome characterized pathologically by degeneration of the frontal and temporal lobes and clinically by deficits in behavior, executive function and/or language [9,10].

In the last decades, increasing evidence for the tight relationship between ALS and frontotemporal dementia (FTD) based on clinical, genetic and histopathological aspects has been established [10,11]. Histopathologically, over 95% of ALS patients and about 50% of FTD patients are characterized by the presence of intracellular aggregates of misfolded TDP-43 [12]. Thus, ALS and FTD-TDP43 are considered the two phenotypic extremes of a single clinicopathologic spectrum [10,13].

Genetically, some genes have been described to cause both diseases following a mendelian (usually autosomal dominant) inheritance, such as an hexanucleotide expansion in *C9orf72* or pathogenic sequence variants in the *TBK1, VCP* and *TARDBP* genes. However, other genes are particularly associated with only one disease, such as *SOD1* for ALS or *GRN* and *MAPT* for FTD.

Besides mendelian genes, several genetic risk factors have been described to contribute to the development of these diseases. Genetic risk factor is any variant that, upon interacting with the environment or other genetic variants, confers a greater susceptibility for a disease [14].

Expansions of CAG repeats in the first exon of *ATXN2* gene were described as an ALS genetic risk factor over a decade ago [15]. This abnormal CAG expansion had been previously described as the causative molecular mechanism of spinocerebellar ataxia type 2 (SCA2) [16,17], a neurodegenerative disease characterized by uncoordinated movements, postural tremor, decreased muscle tone, weakness and cognitive impairment [18,19]. *ATXN2* is localized on chromosome 12 (12q23–24) and comprises 25 exons. The protein encoded by this gene is named ataxin-2. This protein takes part in several cellular processes like RNA metabolism, endocytosis, cytoskeleton reorganization and calcium-mediated signaling, among others [18,19].

Alleles below or equal to 31 repeats are present in normal individuals [20]. In contrast, 33 and 34 CAG repeats are related to reduced-penetrance SCA2, whereas alleles with a repeat number >34 are considered full-penetrant for SCA2, presenting an autosomal dominant pattern of inheritance [16,17,18,19]. 

CAG-triplet expansion encodes for a polyglutamine region (polyQ) in ataxin-2. This classifies SCA2 within the group of polyQ diseases such as Huntington’s disease [21]. This polyglutamine region is thought to result in a toxic gain of function affecting specific types of neurons such as Purkinje cells, basal ganglia neurons and motor neurons of the brain and spinal cord [19]. In fact, in SCA2, patient motor neurons are known to degenerate [22]. These data suggest that *ATXN2* could be a suitable candidate as a genetic risk factor for ALS. 

Since the initial description, the association between an intermediate CAG repeat expansion in *ATXN2* and an increased risk for ALS has been replicated in several studies, but the threshold for increased risk ranges from 27 to 33 repeats [15,22,23,24,25,26,27]. It is unknown if the variations in the reported expansion lengths could be due to the genetic background of the population [28,29,30]. 

Moreover, only a few studies [27,31,32] have assessed the role of CAG expansions in *ATXN2* as a factor of bad prognosis in ALS patients, with contradictory results [33].

In FTD, the role of ATXN2 intermediate CAG expansions in predisposing to FTD has not been established, but one study suggested that ATXN2 may act as a phenotype modifier [28].

This study aimed to assess *ATXN2* CAG expansions as a genetic risk factor and phenotypic modifier for ALS and FTD in the Spanish population.

## 2. Materials and Methods

### 2.1. Patient Population

ALS and FTD patients were recruited from three referral centers in Spain (12 de Octubre Hospital Research Institute (Madrid), La Fe Hospital (Valencia) and Bellvitge University Hospital (Barcelona). Controls, which included healthy individuals as well as patients with other neuromuscular diseases (congenital myopathies, Steinert’s myotonic dystrophy, familial periodic paralysis non-dystrophic myotonia and spinal muscular atrophy), were recruited in HU12O and HUB. 

ALS patients included in the study were diagnosed following the Gold Coast criteria, where ALS can be diagnosed when both upper and lower motor neuron signs are found in at least one region or isolated lower motor neuron signs are found in two different regions [34]. FTD patients included in this study met the criteria for any of the following variants, which are the phenotypes most commonly associated with motor impairment [35,36]: 108 patients with a behavioral variant frontotemporal dementia (bvFTD), 21 patients with a nonfluent/agrammatic variant (PPA-NF) and 8 patients with a semantic variant (PPA-DS).

### 2.2. Genetic Analysis

The CAG repeat expansion in *ATXN2* and the *C9orf72* GGGGCC repeat expansion were analyzed in all patients and controls in the genetic molecular laboratories of all three hospitals.

#### Determination of CAG Repeat Expansion Length in *ATXN2* and *C9orf72*

DNA from whole peripheral blood was extracted with an Illustra Nucleon Genomic kit (GE Healthcare Life Science). CAG repeats in *ATXN2* and GGGGCC repeats in *C9orf72* were determined by polymerase chain reaction (PCR) followed by capillary electrophoresis. 

The method used for testing *C9orf72* expansion was adapted from the previous literature [37,38].

Standard PCR for *ATXN2*

The primers used in the *ATXN2* gene amplification are shown in Appendix A, and the volumes are shown in Appendix A. The PCR procedure for ATXN2 involved an initial stage of 95 °C for 5 min, followed by 42 cycles (97 °C for 15 s, 53 °C for 20 s and 72 °C for 1 min) and a final stage at 72 °C for 10 min. 

PCR products were diluted to 1/10 and mixed with formamide and GeneScan^TM^ 400HD Rox^TM^ dye size standard. We used a 3130 Genetic Analyzer with a 3130xl Genetic Analyzer Upgrade (Applied Biosystem^TM^, Foster City, CA, USA). The results were analyzed by a Peak Scanner, 2.0 version (ThermoFisher Scientific, Foster City, CA, USA). We used two internal controls of 27 and 37 CAG repeats, which were provided by the immunology service of 12 de Octubre Hospital.

When the PCR product analysis showed two peaks, we assumed those lengths for both alleles. Conversely, if we obtained a single peak, it could have been due to two main reasons: this patient could have carried both alleles with the same length, or one of the alleles was large enough not to be amplified by conventional PCR. Therefore, single-peak samples were amplified by the repeat-primed PCR procedure (see primers in Appendix A).

Repeat-primed PCR for *ATXN2*

The PCR procedure involved an initial stage of 95 °C for 5 min, followed by 30 cycles (94 °C for 30 s, 60 °C for 30 s and 72 °C for 30 s) and a final stage at 72 °C for 10 min. Afterwards, capillary electrophoresis was performed, as detailed above. 

The RP-PCR pattern was presented as a succession of peaks with 3 base pairs of difference corresponding to each of the CAG triplet repeats. The first peak corresponded to 7 repeats (which were included in the forward primer SCA-P4), and each subsequent peak added 1 more repeat. In this way, normal alleles showed a few large peaks, while expanded alleles showed a multitude of peaks, each of them decreasing in size (the PCR conditions are available in Appendix A).

### 2.3. Data Analysis

Data were summarized using the mean and standard deviation for the continuous variables and relative and absolute frequencies for the categorical variables. 

The one-tailed Fisher test was used to evaluate the association between the different intermediate CAG repeat expansion (RE) sizes in *ATXN2* and both ALS and FTD diseases. Subsequently, the relationships between demographic (sex), clinical (type of onset and presence of familial history) and genetic variables were explored with the chi-squared test. 

A means comparison of the age of onset between carriers and non-carriers of an intermediate RE in *ATXN2* was performed using the Mann–Whitney U test. To achieve this, a Kolmogorov–Smirnov test was first performed to check the normality. The impact on survival was evaluated using a Kaplan–Meier curve, the log–rank test, and a Cox regression model, which was adjusting by the following variables: ≥27 repeats, bulbar onset, age of onset, frontotemporal dementia (FTD) and female sex. The CAG intermediate expansion length in ATXN2 was included as a random effect. Statistical analyses were performed using the SPSS Statistics 24 software (IBM, Madrid, Spain), and graphics were created using the Graph Pad Prism 8.0.1 software support.

### 2.4. Ethics Approval

Participants expressed their consent for data collection and genetic studies in their respective centers. Information for all patients was de-identified, and data collection and processing were approved by the Ethics Committee for Biomedical Research of the 12 de Octubre Hospital Research Institute (Madrid), La Fe Hospital (Valencia) and Bellvitge University Hospital (Barcelona). This study was conducted according to the principles expressed in the Declaration of Helsinki. This genetic study was approved by the local CEIm at Hospital 12 de Octubre Research Institute. All patients received a patient information sheet, and an informed consent was obtained from all subjects involved in the study.

## 3. Results

### 3.1. RE in ATXN2

A total of 757 patients (620 ALS and 137 FTD patients) and 362 controls were included in this study. Their demographic, clinical and genetic characteristics are summarized in Table 1. In the ALS patients, the CAG repeat expansion size ranged from 22 to 36 repeats, in the FTD patients, it ranged from 22 to 30 repeats, and in the controls, it ranged from 22 to 33 repeats. Regarding allele frequencies, in the ALS patients, the most common allele was (CAG)_22_ (52.1%), followed by (CAG)_23_ (32.1%). The most frequent alleles in the FTD patients were (CAG)_22_ (48.2%) and (CAG)_23_ (27.0%). Finally, the predominant allele among the control group was also (CAG)_22_ (51.1%), followed by (CAG)_23_ (31.8%) (Figure 1).

### 3.2. RE in ATXN2 as a Risk Factor for ALS

Intermediate repeat alleles of ≥27 CAG repeats were more frequent in the ALS patients than in the controls (9.03% vs. 3.59%) (Figure 1b). This association was statistically significant considering ≥27, ≥28 and ≥29 as intermediate lengths of the *ATXN2* expansion (Table 2).

### 3.3. Clinical Characteristics of ALS Patients with RE in ATXN2 

Fourteen percent of the ALS patients with an intermediate CAG repeat expansion (≥27 CAG) developed ALS–FTD compared to 12.5% of those not carrying the expansion. In addition, 78.85% of them showed limb onset ALS vs. 66.31% of those with less than 27 repetitions, although these differences were not statistically significant. A statistically significant association was found between limb onset and being a carrier of ≥28 repetitions (*p* = 0.0139). Finally, an association was found between being a carrier of ≥28 repetitions and having a positive familial history of ALS (*p* = 0.0009). No differences in age, male/female ratio nor co-mutations were observed (Table 3). 

The mean survival of the ALS patients with intermediate CAG expansion (29.23 ± 12.77) was shorter than that of the ALS patients with normal CAG repeats (40.69 ± 28.13), and this difference was statistically significant (*p* = 0.0458). The Kaplan–Meier curves (Figure 2) and log–rank tests showed a shorter median survival in the patients with ≥27 CAG repeats in the ALS patients than those with shorter lengths (34.5 vs. 29.1 months, *p* = 0.004). In the Cox regression model, being a carrier of ≥27 CAG was an independent factor for poor survival (hazard ratio [HR] 1.74 [1.18, 2.56], *p* = 0.005) together with bulbar onset and age of onset (Table 4 and Figure 3).

The longest survival was observed in the group of patients with limb onset who were non-carriers of an intermediate *ATXN2* expansion (red), followed by the group with bulbar onset who were non-carriers of an intermediate expansion (blue). Among the intermediate-expansion-carrying groups (≥27 CAG repeats), patients with bulbar onset (green) also exhibited shorter survival. 

### 3.4. RE in ATXN2 as a Risk Factor for FTD

We also analyzed the presence of intermediate CAG repeat expansion alleles in the FTD patients and controls. Intermediate repeat expansion alleles (27–33) were numerically more frequent in the FTD patients than in the controls (Figure 1), but this difference was not statistically significant (*p* > 0.05) (Table 5). 

Finally, no statistically significant association was found between age of onset or family history of FTD and being a carrier of intermediate repeats (*p* > 0.05).

### 3.5. Concomitant RE in ATXN2 and Expansions in C9ORF72

Six (10%) of the sixty ALS patients and one of the seven FTD (14.28%) patients carrying a full-length *C9orf72* RE were also carriers of an intermediate expansion in *ATXN2* (Table 1). All the ALS patients carrying co-expansions had a family history of ALS and/or FTD vs. 57% of the ALS patients carrying the *C9orf72* RE only. The mean age of onset of the patients with co-expansions was 55 years, with 16% of the patients showing bulbar onset (16.67%) and a median survival of 36.4 months (data available for only two patients). In comparison, the patients with *C9orf72* RE only had a mean age of onset of 58 years, with 31% of them having a bulbar onset and a median survival of 39 months. The single FTD patient exhibiting co-expansion also had a positive family history.

## 4. Discussion

### 4.1. RE in ATXN2 Gene in ALS Patients

Since Elden and colleagues [15] reported the association between an intermediate CAG repeat expansion (27–33) in *ATXN2* and an increased risk for ALS, several studies have widely reaffirmed this association (Table 6). Even though most studies have focused on populations of European ancestry, there has not been a consensus established in terms of length.

The results of this study carried out in the Spanish population were consistent with previous studies that have proposed an intermediate-length CAG repeat expansion in *ATXN2* as a genetic risk factor for ALS. Specifically, according to our data, the risk of developing ALS started with 27 repeats and increased proportionally with the increase in the expansion range, suggesting a “length–response” risk as follows: ≥27 repeats (OR = 2.666 [1.471–4.882]; *p* = 0.0006), ≥28 repeats ([OR] = 3.021 [1.547–6.309]; *p* = 0.0005) and ≥29 repeats ([OR] = 7.563 [2.098–32.550]; *p* = 0.0005). Overall, this theoretical effect must be taken cautiously considering that the confidence intervals for each repeat expansion largely overlapped. However, this finding is in keeping with a previous meta-analysis that found that the risk increased exponentially with length for alleles of 29–32 repeats [33]. Moreover, a “length–response” effect could explain the fact that the association between *ATXN2* and ALS is more consistently found for ≥29 than for <29 repeats throughout the literature (Table 6).

Considering the multistep hypothesis [43], other genetic and environmental factors could act as risk modifiers of ALS concomitantly with *ATXN2* intermediate REs. A limitation of our study is that only the concomitant effect of *ATXN2* intermediate REs and *C9orf72* was assessed (see below). Thus, larger studies are needed to understand if, besides a “length–response” risk, other factors could explain the different thresholds of CAG repeats found among the studies in the literature, even in populations of European ancestry (Table 6).

Our study confirms the association of an intermediate RE and the presence of family history. This was found previously in some studies [22,39,44], but not in others [41].

Regarding the survival results, the ≥27-repeat ALS patients showed a strikingly reduced survival (29.23 ± 12.77) compared to the ALS patients with non-expanded alleles (40.70 ± 28.13). Our results are consistent with previous reports in the Italian population [27,31] but are opposed to studies conducted in the UK, The Netherlands and USA [32,33]. These differences may be due to the diverse genetic background (Mediterranean vs. north European) or to selection bias in the studied populations (i.e., our cohort was enriched for familial cases). Interestingly, we confirmed here that the effect of the RE in survival was independent of other variables. Moreover, our data suggest that the patients with spinal onset carrying an intermediate RE in *ATXN2* had an even worse prognosis than the patients with bulbar onset not carrying the RE. Thus, this information might be useful when stratifying patients in clinical trials. 

Pathologically, ALS patients with an *ATXN2* intermediate RE show a significant loss in Purkinje cells in the cerebellar vermis [45] and, more remarkably, significantly greater amounts of phosphorylated TDP-43 in the spinal cord [46] than sporadic patients. Elden and colleagues [15] showed that an intermediate RE enhances the interaction of ataxin-2 with TDP-43, making TDP-43 more prone to mis-localize from the nucleus to the cytoplasm under stress situations, thus increasing TDP-43 aggregation and toxicity. These findings could explain the pathogenic role of an *ATXN2* intermediate RE in ALS patients. Interestingly, a recent study [47] reported an increased survival and motor neuron function after lowering ataxin-2 levels with both a transgenic murine model and treatment with antisense oligonucleotides (ASOs). This suggests that lowering ataxin-2 levels has therapeutic potential for ALS, and to prove it, a clinical trial with ASOs directed against *ATXN2* is underway (NCT04494256). 

### 4.2. Association of Intermediate CAG Repeat Expansion in ATXN2 Gene with FTD 

As mentioned above, ALS and FTD are closely associated not just based on histopathology (TDP-43 inclusions) but also in clinical and genetic aspects [10,11]. Despite these similitudes, we were not able to corroborate any effect of *ATXN2* intermediate REs on FTD risk. Specifically, although more FTD patients than controls carried intermediate REs, this difference was not statistically significant. However, it must be highlighted that the sample size of the FTD patients was considerably lower than that of the ALS patients and, thus, our study might be underpowered to detect this association. Other studies, probably underpowered too, also failed to find this association [29,44]. However, larger studies have confirmed that an intermediate RE in *ATXN2* is most frequent among FTD patients than in controls [28,32], although with lower ORs than those found for ALS. This reduced association is not surprising, given that about 50% of FTD patients do not show TDP-43 inclusions. We did not find any differences in age of onset or family history in the FTD patients carrying an intermediate RE. The survival data were largely lacking due to the reduced death rate of the FTD patients compared with that of the ALS patients. Moreover, given the limited sample size, we did not assess the association between an intermediate RE in *ATXN2* and the different phenotypes of FTD (bvFTD, PPA-NF and PPA-DS). Finally, we do not have any data on patients who started with FTD and developed motor neuron signs later in the disease course. Interestingly, another study suggested intermediate alleles in *ATXN2* as phenotypic modifiers in FTD [29], reporting a positive association between an intermediate number of CAG repeats and an earlier onset of the disease, Parkinsonism and psychotic symptoms at disease onset. Larger studies of well-phenotyped FTD patients with pathological confirmation are needed to figure out other associations between specific phenotypic characteristics and *ATXN2* intermediate REs.

### 4.3. Concomitant RE in ATXN2 and Expansions in C9orf72

In our cohort, 10% of the ALS patients and 14.3% of the FTD patients carrying *C9orf72* expansions also harbored an intermediate RE in *ATXN2*. Thus, the frequency of intermediate REs in the ALS patients carrying C9orf72 expansions was similar to that found in the sporadic patients. This is in contrast with two studies that found a lower (0.6–2%) frequency of co-expansions [48,49]. Despite the small sample size, our data suggest that co-expansions are not infrequent in the Spanish population and may act as a modifier factor of the penetrance and phenotype of *C9orf72*. Firstly, 100% of the ALS patients with both expansions vs. only 57% of those with *C9orf72* RE only reported a family history of ALS or FTD. Recently, the penetrance of *C9orf72* has been found to vary between families [50]. Thus, the presence of an *ATXN2* intermediate RE might be one of the factors that determines this variable penetrance. Secondly, bulbar onset was half as frequent in the patients with both expansions than in those with only the *C9orf72* expansion. Thirdly, the patients carrying both expansions showed a slightly earlier disease onset and shorter disease duration than those carrying only the *C9orf72* expansion. These data must be taken with caution, given that only seven patients carried both expansions, but again, the data suggest additive toxic effects of both expansions. Interestingly, a study in zebrafish showed that the loss of *C9orf72* synergizes with polyQ ataxin-2 to induce motor neuron dysfunction and cell death [51]. Further molecular studies should assess this potential interaction.

In summary, this study confirms that intermediate CAG RE in *ATXN2* is a genetic risk factor and a also phenotype modifier in ALS in the Spanish population. 

## 5. Conclusions

Intermediate repeat expansions in *ATXN2* greater or equal to 27 repeats are a genetic risk factor for ALS in the Spanish population, with larger expansions being associated with greater risk.

A statistically significant association between intermediate CAG repeat expansions in the *ATXN2* gene and FTD patients could not be demonstrated in the Spanish population.

In the ALS patients, intermediate repeat expansions in *ATXN2* were associated with positive family history, limb onset and shorter survival compared with the patients without an intermediate expansion.

In patients carrying a *C9orf72* expanded allele, the presence of intermediate repeat expansions in *ATXN2* might influence the penetrance and phenotype of the disease.

## Figures and Tables

**Figure 1 biomedicines-12-00356-f001:**
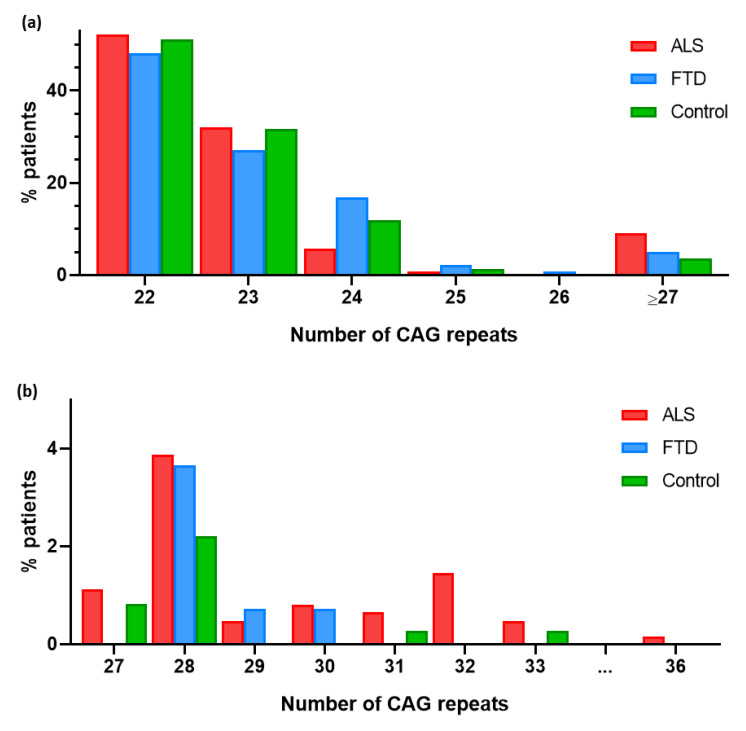
(**a**) Graphical representation of the percentage of patients carrying each CAG expansion allele in *ATXN2* gene. (**b**) Frequency of CAG expansion alleles in *ATXN2* in ALS, FTD and control groups. Close-up view of 27-to-36 region.

**Figure 2 biomedicines-12-00356-f002:**
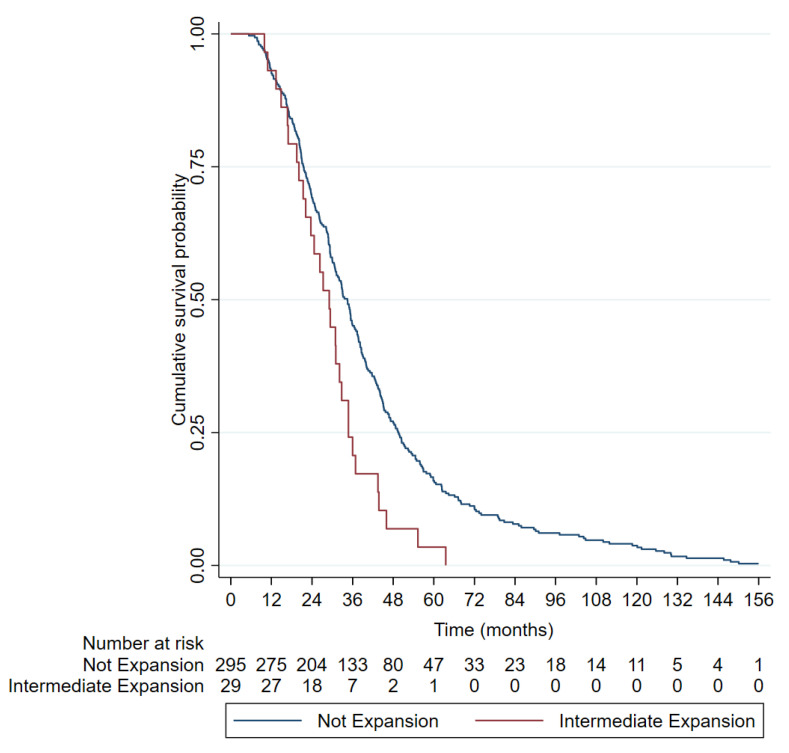
Kaplan–Meier curves representing the survival of amyotrophic lateral sclerosis patients according to number of *ATXN2* repeats (≥ or <27). Survival data were available from 29 out of 56 ALS patients with ≥27 RE and from 295 out of 564 <27 RE.

**Figure 3 biomedicines-12-00356-f003:**
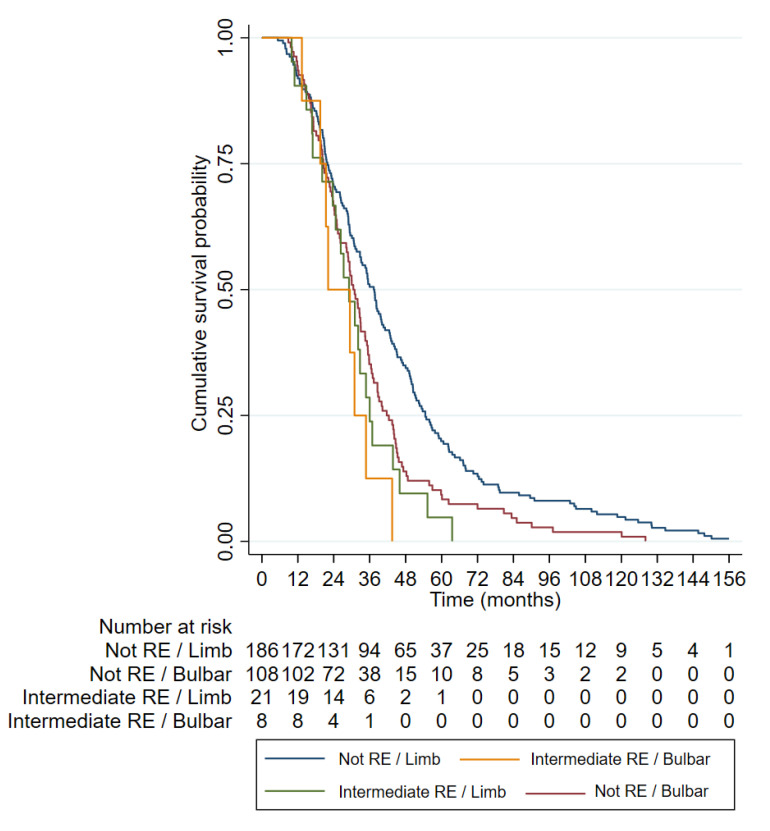
Kaplan–Meier curves representing the categorical variables included in the multivariable Cox regression survival model.

**Table 1 biomedicines-12-00356-t001:** Demographic and clinical characteristics of ALS patients, FTD patients and controls.

Clinical Characteristics	ALS	FTD	Controls
Comorbidity, n (%)	87 (14.03%)	18 (13.14%)	-
Male sex, n (%)	337 (54.35%)	81 (59.12%)	175 (48.34%)
Limb onset, n (%)	415 (66.94%)	-	-
Familial, n (%)	142 (22.90%)	38 (27.74%)	-
*ATXN2* ≥ 27 CAG repeats, n (%)	56 (9.03%)	7 (5.11%)	13 (3.59%)
*C9orfF72* expansion, n (%)	60 (9.68%)	7 (5.11%)	0 (0.00%)
*ATXN2* ≥ 27 CAG repeats and *C9orfF72* expansion, n (%)	6 (10.0%)	1 (14.3%)	0 (0.00%)
Age of onset (years, mean ± SD (range)	59.87 ± 13.45 (16.4–95.6)	63.16 ± 9.85 (39.3–81.8)	56.32 ± 15.64 (18.5–85.8) *
Survival (months, mean ± SD) (range)	39.67 ± 27.30 (5.3–156.1)	99.32 ± 51.90 (21.8–216.0)	-
Total ALS samples	620	137	362

* In control group, age at sample collection.

**Table 2 biomedicines-12-00356-t002:** CAG repeat expansion alleles in *ATXN2* in control group and ALS patients, *p*-value and odds ratio.

CAG Repeats	ALS	Control	*p*-Value ^a^	Odds Ratio ^b^
≥27 CAG repeats	56 (9.03%)	13 (3.59%)	0.0006	2.666 [1.471–4.882]
≥28 CAG repeats	49 (7.90%)	10 (2.76%)	0.0005	3.021 [1.547–6.309]
≥29 CAG repeats	25 (4.03%)	2 (0.55%)	0.0005	7.563 [2.098–32.550]

^a^ Fisher’s test (statistical significance *p* < 0.01); ^b^ 95% confidence interval.

**Table 3 biomedicines-12-00356-t003:** Clinical and genetic characteristics of ALS patients with the different positive cut-offs for RE *ATXN2* gene.

ALS						
	<27	≥27	<28	≥28	<29	≥29
Mean age of onset (years)	59.95	59.16	59.96	58.66	59.81	61.35
Mean survival (months)	40.69	29.23	40.47	29.68	39.90	33.66
*C9orf72* + (n)	54 (9.57%)	6 (10.71%)	54 (9.46%)	6 (12.24%)	58 (9.75%)	2 (8%)
Other genes + (n)	94 (16.66%)	10 (18.52%)	94 (16.46%)	10 (20.41%)	100 (16.81%)	4 (16%)
Male/female (n/n; ratio)	309/254 (1.217)	28/28 (1)	310/260 (1.192)	27/22 (1.227)	324/270 (1.2)	13/12 (1.083)
Limb/bulbar (n/n; ratio)	374/176 (2.125)	41/11 (3.727)	378/179 (2.111)	37/8 (4.625)	396/184 (1.467)	29/3 (6.333)
Family history (+/−; %)	122/442 (21.63%)	20/36 (35.71%)	122/449 (21.37%)	20/29 (40.28%)	131/464 (22.02%)	11/14 (44.00%)
FTD (yes/no; %)	79/485 (14.01%)	7/49 (12.5%)	79/492 (13.84%)	7/42 (14.29%)	83/512 (13.95%)	3/22 (12.00%)

+/− Refers to the number of patients with and without family history, respectively. n/n Refers to the number of males and females and limb and bulbar onset, respectively, within those categories.

**Table 4 biomedicines-12-00356-t004:** Cox’s model assessing the co-variables influencing survival in ALS patients.

	Univariable			Multivariable		
Variable	HR	CI (95%)	*p*-Value	HR	CI (95%)	*p*-Value
≥27 repeats	1.74	1.181–2.563	0.005	1.782	1.209–2.628	0.004
Bulbar onset	1.415	0.538–0.856	0.001	1.400	1.105–1.775	0.004
Age of onset	1.016	1.007–1.026	0.001	1.015	1.005–1.824	0.003
FTD	0.992	0.7748–1.270	0.949	-	-	-
Female sex	1.253	1.003–1.565	0.046	-	-	-

HR: Hazard ratio. CI: Confidence interval.

**Table 5 biomedicines-12-00356-t005:** CAG repeat expansion in *ATXN2* in control group and FTD patients, *p*-value and odds ratio.

CAG Repeats	FTD	Control	*p*-Value ^a^	Odds Ratio ^b^
≥27 CAG repeats	7 (5.11%)	13 (3.59%)	0.220	1.446 [0.558–3.574]
≥28 CAG repeats	7 (5.11%)	10 (2.76%)	0.099	1.895 [0.7295–5.241]
≥29 CAG repeats	2 (1.46%)	2 (0.55%)	0.155	2.667 [0.4134–17.12]

^a^ Fisher’s test, ^b^ 95% confidence interval.

**Table 6 biomedicines-12-00356-t006:** Intermediate CAG repeat expansion in *ATXN2* gene associated with ALS extracted from 12 studies with statistical determinations.

Study	CAG Repeat Number	*p*-Value ^a^	Odds Ratio ^b^
North America [15]	27–33	0.000036	2.8 [1.54–5.12]
Europe [22]	>30	0.0062	-
France and Canada [39]	≥29	0.00024	5.5 [1.9–15.9]
Europe [40]	>30	0.004	5.74 [4.26–7.22]
Flanders [23]	27–33	0.012	-
South Italy [41]	≥28	0.001	5.832 [1.71–9.78]
Turkey [25]	>30	0.01721	-
Meta-analysis [24]	29–33	-	>1
Europe and North America [26]	30–33	0.0001	4.44 [2.91–6.76]
Sardinia [27]	≥31	0.0001	-
Brazil [42]	≥26	0.005	2.56 [1.29–5.08]
Europe [32]	≥31	9.50 × 10^−7^	6.93 [3.19–15.02]
Spain (present study)	≥27	0.0013	2.666 [1.471–4.882]

Statistical determinations include *p*-value and odds ratio (if present). ^a^ Fisher’s test, ^b^ 95% confidence interval.

## Data Availability

Anonymized data not published within this article will be made available by request from any qualified investigator.

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
