# Peer review of "Intermediate Repeat Expansion in the ATXN2 Gene as a Risk Factor in the ALS and FTD Spanish Population"

_biomedicines, 2024, doi:10.3390/biomedicines12020356_

Round 1
Reviewer 1 Report
Comments and Suggestions for Authors
This is an interesting report describing the frequency and prognostic role of ATX2 intermediate expansions in a multicenter Spanish cohort of ALS and FTD patients. Results are nicely discussed using a critical approach. However, some limitations should be better acknowledged, and missing information should be provided to evaluate the overall significance and soundness of the authors' findings.
- First of all, important information regarding the diagnostic criteria used for patient selection are currently missing. It is also unclear whether the authors include in the FTD label only patients with behavioral variant of FTD or also primary progressive aphasia patients. Also, ALS-FTD patients were considered as ALS or FTD for subgroup analyses?
- Based on Table 3, the term "cognitive impairment" was used as synonym of FTD. This is incorrect, as cognitive and/or behavioral impairment are labels that are used for ALS patients with deficits that do not configure a full-blown diagnosis of dementia. Please, specify.
- In the methods section, it is unclear which variables were included in the Cox regression models. This should be specified.
- The authors did not assess the presence of genetic pathogenic variants other than ATXN2 and C9orf72 expansions. This limitation should be clearly acknowledged.
-As a minor comment, "In FTD, one study suggested that ATXN2 may act as a phenotype modifier" is followed by 2 references, therefore it unclear which one the authors are referring to.
Comments on the Quality of English LanguageI detected some minor typos or inaccuracies (e.g., U "de" Mann-Whitney test, some commas not appropriately placed).
Reviewer 2 Report
Comments and Suggestions for Authors
This study investigates the impact of intermediate CAG expansions in the ataxin-2 (ATXN2) gene on the risk and phenotype of amyotrophic lateral sclerosis (ALS) and frontotemporal dementia (FTD) in the Spanish population. The results reveal that ≥27 CAG repeats in ATXN2 are associated with a higher risk of developing ALS but not FTD. ALS patients with ≥27 repeats exhibit shorter survival rates, more frequent limb onset, and a higher family history of ALS. The study suggests a potential modifying role of ATXN2 expansions in patients with C9orf72 expansions.
1. The authors should explain the potential variations in the threshold for increased risk within the intermediate CAG expansion range?
2. How might the genetic background of different populations influence the variations in the reported expansion lengths?
3. Are there specific characteristics or biomarkers that distinguish ALS patients with intermediate ATXN2 expansions from those without in terms of disease progression?
4. The study mentions the potential modifying role of ATXN2 expansions in patients with C9orf72 expansions. Could this interaction be further elucidated with additional molecular or functional analyses?
5. Given the clinical overlap between ALS and FTD, could the study delve into more detailed phenotypic characteristics to identify potential subtle connections between ATXN2 expansions and FTD?
Reviewer 3 Report
Comments and Suggestions for Authors
The authors present an interesting study evaluating the potential role of ataxin-2 gene as a risk factor for ALS, especially in the intermediate expansion group. Patients with full expansion have been classically associated with Spinocereebellar ataxia type 2 (SCA2). Although not being a completely original finding, as intermediate expansion patients have also been linked both to sporadic ALS and familial ALS type 13, this manuscript brings important contribution to the literature, as it also evaluates the potential risk for FTD, which was negative. Furthermore, the study also discloses the potential occurrence of two genetic mechanisms in the pathophysiology of ALS, which has been also identified in several cases in the literature, but in this case with two repeat expansion disorders (c9orf72, ATXN2). Figures and tables have been properly presented by the authors.
Author Response
We truly thank the reviewer for his/her thorough review and comments.
Round 2
Reviewer 1 Report
Comments and Suggestions for Authors
The authors should clearly declare how many patients had bvFTD and how many had nfvPPA and svPPA variants. It is known that most nfvPPA patients actually do NOT show TDP-43 pathology, as incorrectly declared in the manuscript. Apart from that, I have no further concerns.
Author Response
We truly thank the reviewer for his/her thorough review and comments again.
We have modified the following paragraph to specify the number of cases per clinical subtype of DFT.
ALS patients included in the study were diagnosed following the Gold Coast criteria, where ALS can be diagnosed when both upper and lower motor neuron signs are found in at least one region, or isolated lower motor neuron signs are found in two different regions [34]. FTD patients included in this study met criteria for any of the following variants, which are the phenotypes most common associated with motor impairment [35], [36]: 108 patients with behavioural variant frontotemporal dementia (bvFTD), 21 patients with nonfluent/agrammatic variant (PPA-NF) and 8 patients with semantic variant (PPA-DS).
Thanks again.